# Negative Contrast: A Simple and Efficient Image Augmentation Method in Crop Disease Classification

**Jiqing Li** , **Zhendong Yin \*, Dasen Li and Yanlong Zhao**

School of Electronics and Information Engineering, Harbin Institute of Technology, Harbin 150001, China;
20b905055@stu.hit.edu.cn (J.L.); sen09024@hit.edu.cn (D.L.); zhaoyanlong@hit.edu.cn (Y.Z.)
\* Correspondence: yinzhendong@hit.edu.cn

**Abstract:** Crop disease classification constitutes a significant and longstanding challenge in the domain of agricultural and forestry sciences. Frequently, there is an insufficient number of samples to accurately discern the distribution of real-world instances. Leveraging the full potential of the available data is the genesis of our approach. To address this issue, we propose a supervised image augmentation technique—Negative Contrast. This method employs contrast images of existing disease samples, devoid of disease areas, as negative samples for image augmentation, particularly when the samples are relatively scarce. Numerous experiments demonstrate that the employment of this augmentation method enhances the disease classification performance of several classical models across four crops—rice, wheat, corn, and soybean, with an accuracy improvement reaching up to 30.8%. Furthermore, the comparative analysis of attentional heatmaps reveals that models utilizing negative contrast focus more accurately and intensely on the disease regions of interest, thereby exhibiting superior generalization capabilities in real-world crop disease classification.

**Keywords:** crop disease classification; crop disease dataset; image augmentation

## 1. Introduction

Ever since AlexNet [1] first harnessed deep learning to win the ImageNet [2] competition in 2012, deep-learning-based methodologies have decisively outperformed conventional feature extraction algorithms, such as SIFT [3] and HOG [4]. The exponential development of Convolutional Neural Networks (CNNs) has charted a new course for the crop disease classification problem, as witnessed in an array of diverse studies. For instance, ref. [5] utilized CNNs for rice disease image classification, achieving an impressive accuracy of 95.48%. Ref. [6] employed deep CNN models for four disease classification tasks on cucumber, obtaining a testing set accuracy of 93.4% for the first time. Ref. [7] applied a deep learning model named DenseNet along with fine-tuning techniques to the PlantVillage [8] dataset, thus escalating the accuracy to 99.75%. Ref. [9] incorporated CBAM modules into the ResNet [10] model and examined the impact of embedding location on classification accuracy, achieving a 97.59% accuracy on a common dataset comprising PlantVillage and actual images. Ref. [11] proposed a novel CNN architecture that enabled their model to obtain a cross-validation accuracy of 99.58% using a more intricate and comprehensive dataset of 27 diseases across six crops. Many existing approaches in the realm of crop disease classification [5,9,11–14] typically use images from the same dataset as both training and validation sets, assessing the model's performance via cross validation.

These accurate results would not have been possible without high-quality datasets, the most ubiquitous of which is PlantVillage. This dataset houses 54,303 images of diseased crop leaves for image classification. However, the majority of these images are captured in a laboratory setting, diverging considerably from real-world disease scenarios. Therefore, some recent works [9,11] have opted for a combination of the public PlantVillage dataset and a private dataset for training to enhance the model's performance. Another crop disease

dataset, PlantDoc [15], includes 13 plant species and 17 types of diseases with 2598 samples and is relatively closer to reality. However, this dataset does not cover numerous plant and disease species, necessitating further provision of samples for more diverse plant diseases. Selected images from PlantVillage and PlantDoc are shown in Figure 1. In order to supplement the dataset with more realistic crop disease images, we introduce a new dataset, Plant Real-World, encompassing four crops with a total of 16 diseases.

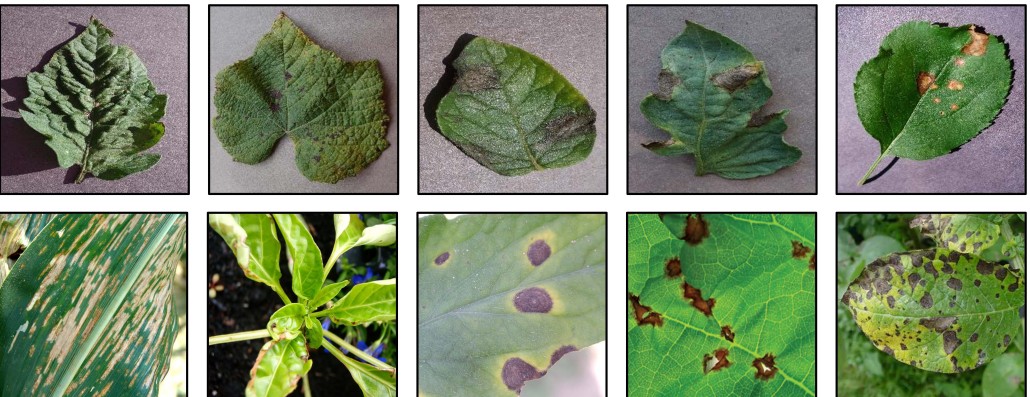

**Figure 1.** The first and second rows are samples from the PlantVillage and PlantDoc datasets, respectively.

Beyond dataset enrichment, deploying appropriate image augmentation techniques can significantly enhance the model's data utilization and, by extension, its performance. Deep learning image augmentation methods can be categorized into three types, namely, model-free, model-based, and optimizing policy-based [16]. Model-free image augmentation methods include Cutout [17], Random Erasing [18], Grid Mask [19], Mixup [20], Cutmix [21], Co-mixup [22], Hide-and-seek [23], and so on. Each method offers a surplus of techniques for single-image or multiple-image augmentation, including geometric transformations, linear fusion, spatial fusion, and other complex transformations, among others. Additionally, there are augmentation methods like [24,25] with GAN [26] as the core, and methods based on optimizing policy exist.

These general methods also prove effective in the field of crop disease classification. Furthermore, our experiments reveal that intelligent use of healthy crop samples can facilitate performance improvements in the crop disease classification task. We term this strategy of utilizing healthy crop samples to enhance accuracy Health Augmentation.

However, the crop disease classification task occasionally includes extremely uncommon diseases, represented by sparse samples of that disease type. This phenomenon is evident across various crop disease datasets, posing a challenge to the model's real-world performance. To tackle this issue, apart from deploying the abovementioned image augmentation methods, we propose a new augmentation technique named Negative Contrast. Using the Adobe PhotoShop CS6, we subtract disease areas from all disease samples in the dataset, then duplicate these processed disease samples as pseudo healthy samples in health augmentation. Through the application of the health augmentation method for training, we manage to significantly improve the performance in crop disease classification tasks under sparse sample conditions.

Specifically, our contributions are:

- We present a compact and realistic dataset, Plant Real-World, for four crops, complete with training and testing sets.
- We introduce a strategy, health augmentation, that leverages healthy crop samples to enhance the performance of crop disease classification. This approach uses healthy crop samples as the negative sample input while making minor modifications to the softmax layer of the network, thereby considerably enhancing the recognition accuracy.

- Building on health augmentation, we further augment the model's generalization performance by using disease samples from which diseased regions have been artificially removed as pseudo healthy samples. With a relatively small training set (5–20% of the original sample count), we obtained an average accuracy improvement of 30.8% across models.

## 2. Materials and Methods

### 2.1. A New Dataset: Plant Real-World

The traditional practice during the training process involves utilizing a fraction of the training set as the testing set in each iteration. By monitoring the trend of accuracy changes along with the increasing number of iterations, we can determine the effectiveness of the model's learning. Finally, the highest accuracy attained during these iterations represents the peak performance of the model.

In our study, we adhered to this same training paradigm. However, our experimental findings revealed that models trained on samples from a homogeneously distributed large dataset (i.e., samples from a similar background or possessing similar discriminative features) exhibited high accuracy during training (occasionally nearing 100%). Yet, the accuracy dropped considerably when tested with real-world crop diseases. Our experiments further showed that these models demonstrated high sensitivity to these backgrounds. We validated this after recombining the images using simple excisions, a situation that is detrimental to the model's generalization performance, as demonstrated in Figure 2. Based on these observations, we believe that providing a dataset with substantial inconsistency between the training and testing sets would offer a more realistic reflection of the model performance.

At present, there is a dearth of crop disease datasets. Previous studies have commonly employed the PlantVillage dataset. To enhance the diversity of the dataset, and to include the most widespread crops from various regions worldwide, we manually curated a collection of images featuring rice, wheat, corn, and soybeans to form a compact but meticulously compiled dataset, namely, Plant Real-World. The training set comprises images obtained from the internet and the PlantDoc dataset, whereas the testing set is assembled from internet-sourced images, disease manuals, and real images. Our testing results, obtained using deep learning models on this dataset, indicated that it is more challenging to identify diseases than using the previous PlantVillage and PlantDoc datasets. This added complexity allows researchers to utilize this dataset to better analyze the performance of deep learning models in crop disease detection tasks.

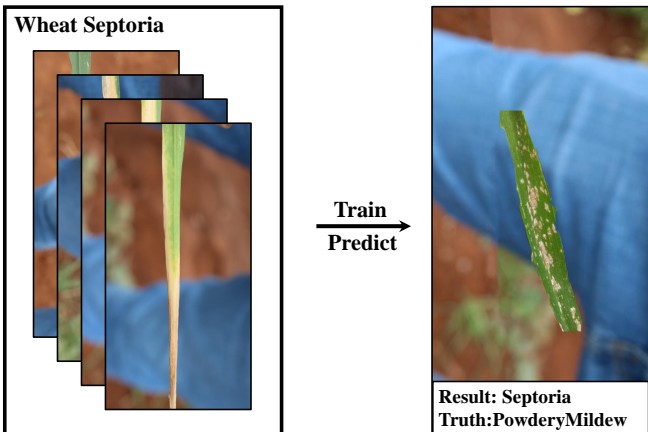

**Figure 2.** The septoria disease background in the wheat training set contained the researchers' blue jeans, which directly led to the fact that the model would give high discriminatory weight to the blue jeans and not enough attention to the powdery mildew region, leading to misclassification.

Therefore, we posit that the use of a real-world test set during the iterative process can offer a more accurate representation of a model's performance. We have compiled a diverse array of disease images sourced from the internet to form a real-world test set, referred to as Plant Real-World. The statistics for each disease category in the Plant Real-World are detailed in Table 1, and a selection of test images from Plant Real-World is illustrated in Figure 3. To maximize the utility of images in each testing set, we refrained from imposing a constant on the ratio of the images in the training set to that in the testing set.

**Table 1.** The statistics of Plant Real-World.

| Crops | Diseases | Training | Testing |
|---|---|---|---|
| Wheat | Flag Smut | 14 | 6 |
| | Mildew | 7 | 1 |
| | Powdery Mildew | 21 | 35 |
| | Septoria | 97 | 29 |
| | Stipe Rust | 207 | 99 |
| Corn | Blight | 75 | 32 |
| | Common Rust | 70 | 17 |
| | Gray Leaf Spot | 58 | 57 |
| Soybean | Downy Mildew | 124 | 26 |
| | Frogeye | 140 | 30 |
| | Septoria | 104 | 30 |
| Rice | Blast | 27 | 25 |
| | Brown Spot | 75 | 21 |
| | Leaf Scald | 53 | 13 |
| | Sheath Blight | 64 | 9 |
| | Tungro | 46 | 8 |

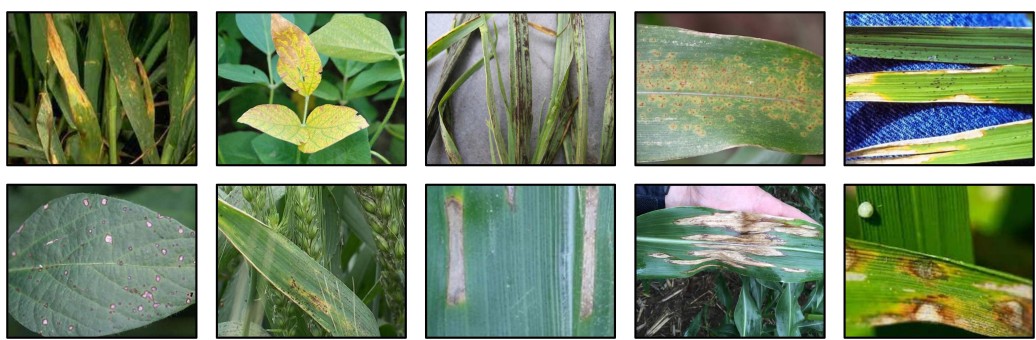

**Figure 3.** Examples of Plant Real-World.

As can be seen from Table 1, the number of specimens of certain diseases in the Plant Real-World section already meets the paradigm of few-shot learning. However, unlike some common few-shot datasets Birds [27] and Stanford Dogs [28], our training set size is also scarce, hence we are hardly able to utilize models such as the Prototypical Network [29] for dataset learning, which is a distinctive feature of Plant Real-World.

Meanwhile, it is not difficult to see from the images that the difficulty of identifying diseases in the field of disease recognition is comparable to that of fine-grained visual classification. The identification of some diseases even requires attention to details such as the texture around the spots, the size of the spotted area, or the density of the spots. Some existing methods [30,31] have been researched for problems like fine-grained visual classification, therefore our dataset can be expanded to this field.

## 2.2. Health Augmentation

Currently, Convolutional Neural Networks (CNNs) are undoubtedly the preferred choice for image feature extraction and classification tasks. As specialized neural networks, CNNs are adept at processing grid-like data, akin to an image constituted of a pixel grid. The design concept of CNNs takes inspiration from the organization of the animal visual cortex, wherein individual neurons are systematically arranged to tile the visual field. The critical element of a CNN is its convolutional layer, capable of automatically and adaptively learning hierarchical spatial features. This convolutional layer contains several filters (or kernels), which traverse the input data, performing dot products between the filter entries and the input to generate a feature map. Additionally, CNNs incorporate pooling layers that reduce the data dimensions by merging the outputs of neuron clusters at one layer into a singular neuron in the subsequent layer. Fully connected layers, connecting every neuron in one layer to every neuron in the subsequent layer, primarily perform pattern classification at the network's end. In this study, we propose a plug-and-play method "Health Augmentation", aimed at improving the performance of the common model ResNet50 [10], as well as two popular lightweight models, ShuffleNetV2 [32] and MobileNetV2 [33]. We then compare the results of the post-training tests on the Plant Real-World dataset for the models before and after using health augmentation.

In the training process for image classification tasks, background information is often perceived as an irrelevant disturbance. When images in the trained classes share high similarities, models may unintentionally utilize background information as a classification basis. One direct approach to alleviate this issue involves increasing the sample size, thus diversifying the backgrounds in the training set to boost the generalization performance. However, this solution is frequently untenable due to cost and difficulty constraints. In instances where expanding the sample size is infeasible, we propose utilizing healthy crop samples from the PlantVillage dataset or equivalent sources.

For certain existing datasets, healthy samples are classified as a single category. However, for crop disease classification tasks, we discovered through experimentation that treating healthy samples as an individual category could potentially hinder disease classification performance. For instance, a healthy leaf with a few minor spots may be wrongly classified by the model as "healthy" rather than "diseased", which contradicts the objective of disease classification tasks—accurate disease categorization. Consequently, the "healthy" category should not be treated identically to regular disease categories. In our approach, we present a novel method of encoding healthy samples, as depicted in Figure 4.

| Class | Septoria | | DownyMildew | | Frogeye | | Health Augmentation | |
|---|---|---|---|---|---|---|---|---|
| Image | 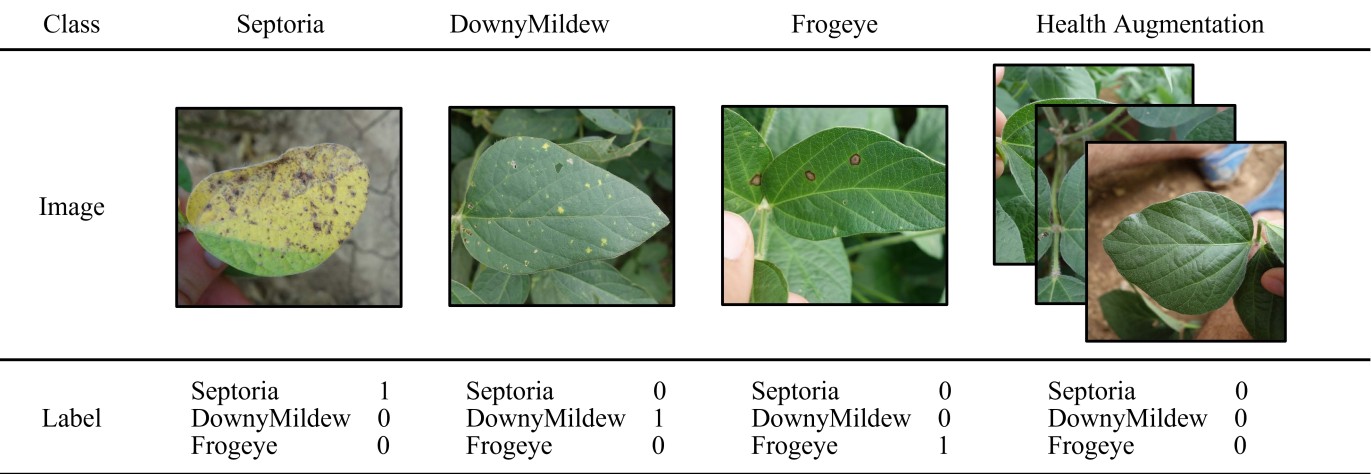 | | | | | | | |
| Label | Septoria | 1 | Septoria | 0 | Septoria | 0 | Septoria | 0 |
| | DownyMildew | 0 | DownyMildew | 1 | DownyMildew | 0 | DownyMildew | 0 |
| | Frogeye | 0 | Frogeye | 0 | Frogeye | 1 | Frogeye | 0 |

**Figure 4.** Coding method for health samples in health augmentation.

Note that we could reevaluate the steps in deep learning image classification. For instance, the ResNet50 image classification network employs a multilayer convolutional kernel as a feature extractor, designed to extract features from images. This is then linked

to a fully connected layer for classification, finally yielding the corresponding probability for each target class after normalization through the softmax layer.

The structure of the softmax layer is illustrated in Figure 5. However, this kind of probability imposes two stringent constraints: first, the probability must fall between 0 and 1; second, the sum of all probabilities must equal 1. Such a definition precludes the possibility of the network producing an all-zero output and compels the network to provide a maximum probability result. We speculate that such a structure may hinder the network's ability to learn from healthy samples, thus constraining its potential learning capacity and necessitating adjustment.

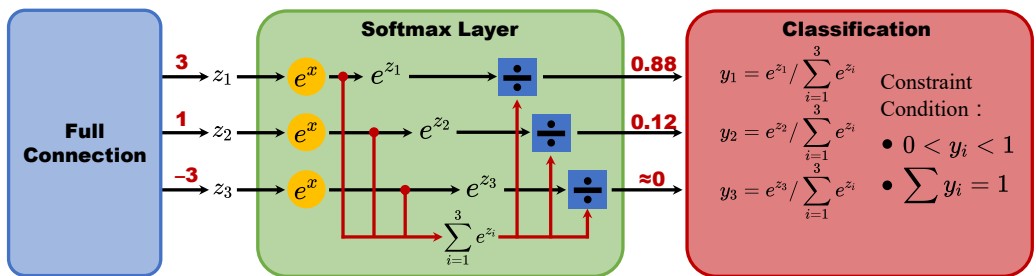

**Figure 5.** The structure and limitations of the softmax layer.

Given these two points, we propose our approach, which we call health augmentation. Our adjustment is a total of two steps.

(1)    Remove the softmax layer.
(2)    Change the loss funciton from cross entropy to mean square error.

When we finish executing step 1, we need to adjust the loss function at the same time, which is step 2. The most common loss function used for multiclassification tasks is crossentropy, which is expressed by the following formula:

$$L = \sum_{i=1}^{N} \boldsymbol{y}_i log(f(\boldsymbol{x}_i)). \tag{1}$$

Since the softmax layer layer is removed, the output value domain of each neuron in the last layer changes from (0,1) to $\boldsymbol{R}$. This makes it easy for the cross-entropy loss function to fail to transmit the error forward. The reason for this is that the true number of the log function is not allowed to be less than or equal to 0, and thus the cross-entropy loss function cannot calculate the error of the back propagation. Therefore, we adjusted the expression of the loss function using the mean square error

$$L = \frac{1}{N} \sum_{i=1}^{N} (\boldsymbol{y}_i - \boldsymbol{x}_i)^2, \tag{2}$$

so that the network without the softmax layer can also work properly.

*2.3. Negative Contrast*

In crop disease classification tasks with sparse samples, Lee [34] endeavored to enhance the classification results by training on ImageNet and subsequently performing transfer learning on the current task. Zhu [35] highlighted that the distribution of the number of object classes in an image conforms to Zipf's law, which posits that the frequencies of certain events are inversely proportional to their rank. This issue of the highly unbalanced distribution of disease samples is also prominent in the field of crop disease classification. For instance, in the test set of Plant Real-World, there are merely seven samples of mildew for wheat disease, in stark contrast to 207 samples of stripe rust. For these particularly scarce samples, training frequently results in overfitting, thus undermining performance

in real-world applications. Therefore, the efficient utilization of sparse data presents a compelling question.

In response to this, we introduce a novel image augmentation method termed negative contrast (hereafter, referred to as NC). This method involves subtracting diseased areas and using content-aware filling of PhotoShop to generate "pseudo healthy samples" for training, with corresponding adjustments made to the labels of these pseudo healthy samples. The complete implementation workflow is depicted in Figure 6. The most comparable image augmentation method to ours is Cut, Paste, and Learn [36]. However, the principal distinction between our method and cut, paste, and learn lies in the fact that the diseased areas we select are not directly merged with different backgrounds for training. Instead, they are generated as "pseudo healthy samples" for training. Building on the altered network structure of health augmentation, the NC employs pseudo healthy samples to expand the training set. We provide a detailed depiction of the inference process of a deep learning model using the NC technique in Figure 7.

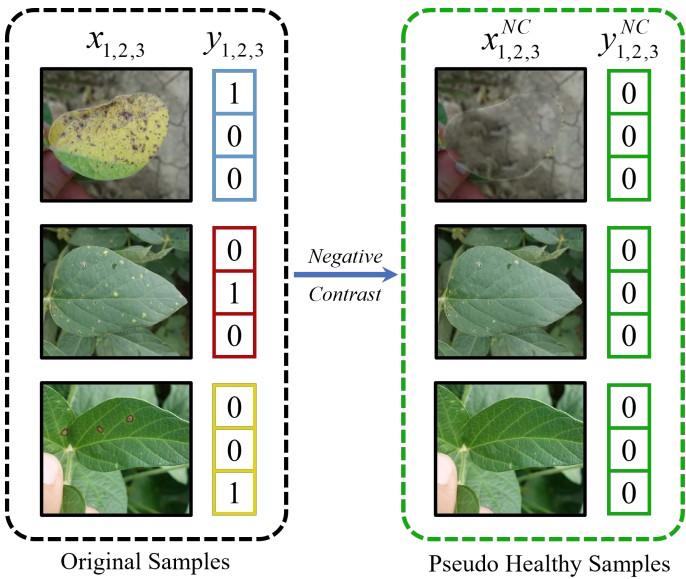

**Figure 6.** Generating pseudo healthy samples using negative contrast.

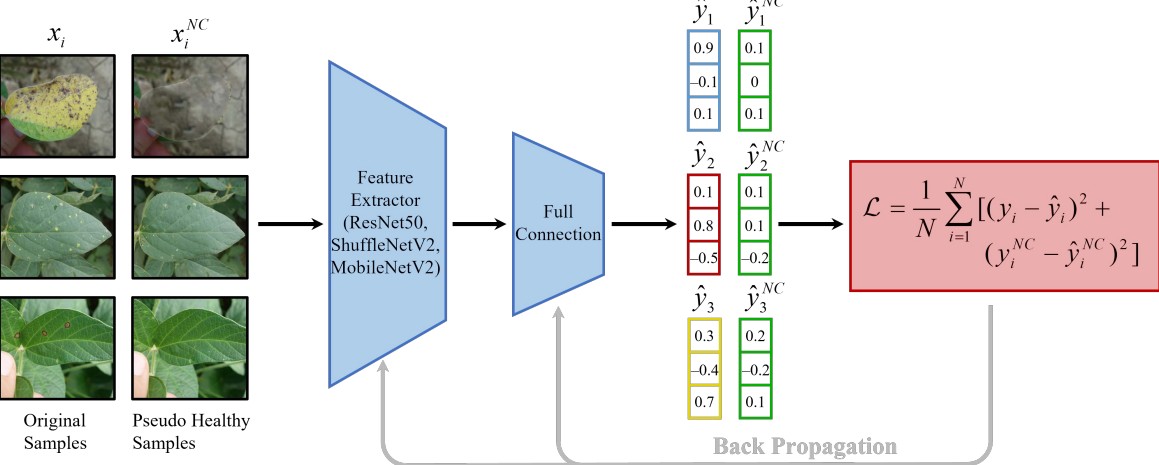

**Figure 7.** Training pipeline of negative contrast.

We discovered that the pseudo healthy samples generated by this method serve the same function as natural healthy crop samples. Both can be incorporated as healthy samples and undergo training using the exact settings employed in health augmentation. This approach can function as a data augmentation method for scarce samples when

acquiring sufficient quantities of healthy samples also proves challenging. This method requires only minimal manual processing to yield a new training set containing twice the amount of original data. We partitioned the dataset with various proportions allocated to the training set, the smallest of which constituted merely five percent of the original, with only five to seven images per category. Despite this limited sample size, the NC was capable of securing further improvements in test accuracy compared to the model utilizing health augmentation. To ensure the comparability of the experiments, we employed an identical number of health samples in the NC and health augmentation.

## 3. Results

### 3.1. Health Augmentation

We implemented health augmentation with MobileNetV2, ResNet50, and ShuffleNetV2 for training and testing, leveraging the Plant Real-World dataset. The experimental outcomes are depicted in Table 2. The results indicate that health augmentation can harness healthy samples to further enhance the precision of crop disease classification. The performance of health augmentation has exhibited consistency across varying models and crops.

**Table 2.** Test accuracy of health augmentation on Plant Real-World.

| Crops | ResNet50 | | MobileNetV2 | | ShuffleNetV2 | |
|---|---|---|---|---|---|---|
| | **Baseline** | **Health Aug** | **Baseline** | **Health Aug** | **Baseline** | **Health Aug** |
| Wheat | 65.9 | 71.2 | 65.3 | 70.0 | 71.2 | 72.9 |
| Corn | 50.9 | 83.0 | 45.3 | 78.3 | 50.9 | 85.9 |
| Soybean | 84.9 | 87.2 | 81.4 | 80.2 | 86.0 | 87.2 |
| Rice | 32.9 | 48.7 | 34.2 | 46.1 | 39.5 | 43.4 |

Health Aug = "Health Augmentation".

### 3.2. Negative Contrast

The experimental results on sparse samples from the Plant Real-World dataset are presented in Table 3. As an image augmentation technique, we compared the performance of negative contrast (NC) with six commonly used image augmentation methodologies. Furthermore, we evaluated the effectiveness of both health augmentation and negative contrast on a dataset reduced to five percent of the original size, using crop disease samples from soybeans. The results from the improvements of the three distinct models demonstrate the superior performance of negative contrast in the task of crop disease detection. The variation in the recognition progress correlated with the number of iterations is depicted in Figure 8.

We carried out image augmentation on the original images utilizing each of these six methods independently and then tested the models trained on these transformed datasets on Plant Real-World. The average accuracy of the three models (ResNet50, ShuffleNetV2, and MobileNetV2) are enumerated in Table 4.

**Table 3.** Test accuracy of negative contrast on Plant Real-World.

| Crops | ResNet50 | | | MobileNetV2 | | | ShuffleNetV2 | | |
|---|---|---|---|---|---|---|---|---|---|
| | **Baseline** | **Health Aug** | **NC** | **Baseline** | **Health Aug** | **NC** | **Baseline** | **Health Aug** | **NC** |
| Wheat (10%) | 63.4 | 62.5 | 67.4 | 59.1 | 60.0 | 73.4 | 63.4 | 66.9 | 71.6 |
| Corn (20%) | 40.7 | 43.4 | 52.8 | 38.8 | 38.8 | 46.2 | 45.9 | 45.9 | 53.8 |
| Soybean (5%) | 59.7 | 69.2 | 84.3 | 41.3 | 53.6 | 72.1 | 68.2 | 74.9 | 78.8 |
| Rice (10%) | 19.8 | 44.6 | 47.2 | 33.5 | 33.7 | 47.2 | 32.4 | 34.1 | 42.5 |

NC = "Negative Contrast".

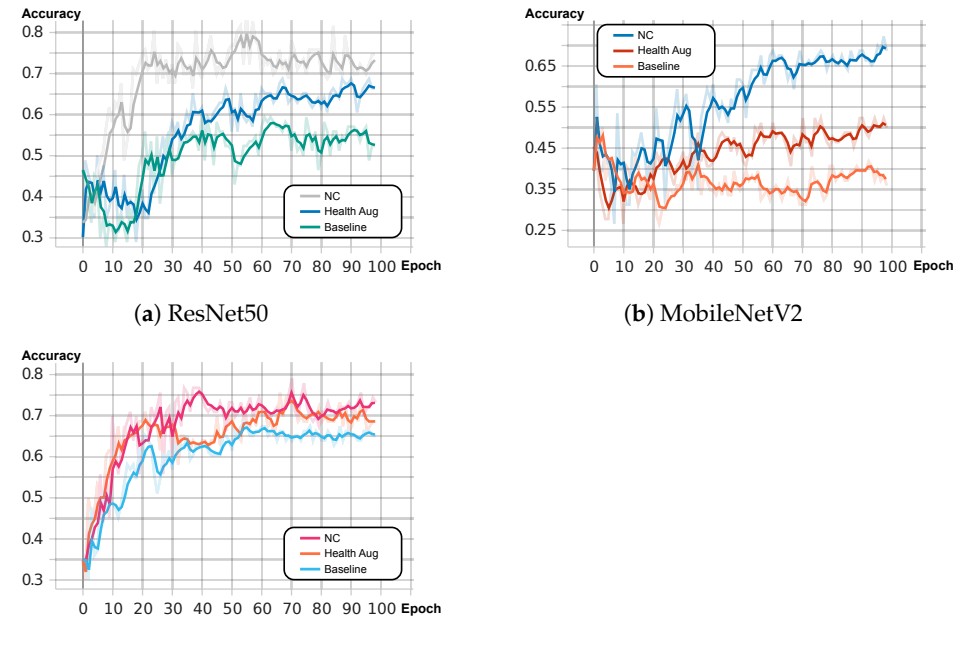

(**a**) ResNet50  (**b**) MobileNetV2

(**c**) ShuffleNetV2

**Figure 8.** Comparison of the performance of the three models before and after improvement.

**Table 4.** The average accuracy of three models using different augmentation methods on Plant Real-World.

| Crops | Baseline | Zoom | Rotate | Color | Brightness | Contrast | Erasing | NC |
|---|---|---|---|---|---|---|---|---|
| Wheat (10%) | 62.2 | 62.4 | 62.2 | 61.0 | 63.3 | 60.8 | 64.7 | 70.8 |
| Corn (20%) | 48.1 | 46.9 | 48.1 | 42.8 | 49.7 | 49.1 | 49.1 | 50.9 |
| Soybean (5%) | 61.2 | 63.6 | 61.2 | 55.8 | 61.2 | 64.7 | 61.6 | 78.4 |
| Rice (10%) | 36.8 | 36.4 | 34.2 | 40.8 | 35.1 | 33.8 | 32.9 | 45.6 |

To delve deeper into the precision of health augmentation and negative contrast concerning the disease's region of interest, we employed the Grad-CAM [37] tool to perform an attentional heat map analysis on these three types of models trained on the Plant Real-World dataset. The methodology is outlined as follows:

To obtain the class-discriminative attentional heat map Grad-CAM $L^c_{Grad-CAM} \in \mathbb{R}^{u \times v}$ in general architectures, the first step is to compute the gradient of $y^c$ with respect to the feature maps $A$ of a convolutional layer, i.e., $\frac{\partial y^c}{\partial A^k_{ij}}$. These gradients are pivotal in obtaining the weights that signify the importance of feature map $k$ for a target class $c$:

$$\alpha^c_k = \frac{1}{Z} \sum_i \sum_j \frac{\partial y^c}{\partial A^k_{ij}}. \tag{3}$$

A linear combination of feature map $A^k$ and weight $\alpha^c_k$ using the RELU activation function will give us our heat map:

$$L^c_{Grad-CAM} = RELU\left( \sum_K \alpha^c_k A^k \right). \tag{4}$$

The heat map illustrated in Figure 9 suggests that compared to the models trained using standard and health augmentation methodologies, the models trained using the negative contrast (NC) approach exhibit stronger attention towards the diseased area while minimizing focus on irrelevant surroundings and healthy disease-free leaves. This provides an intuitive explanation for the enhanced accuracy of models tested with the NC.

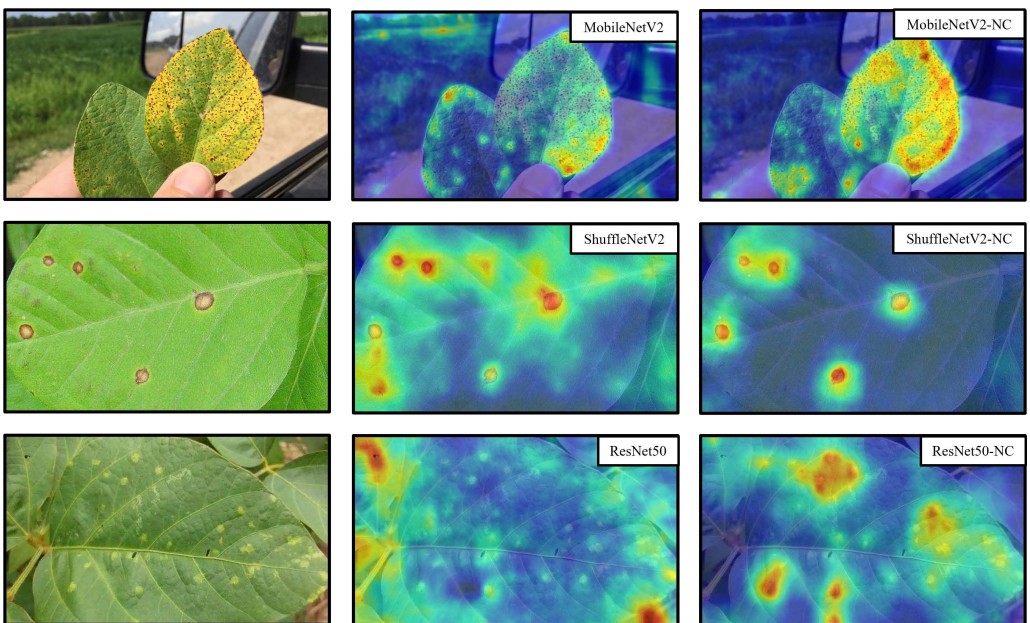

**Figure 9.** Grad-CAM heat map of the three models on Plant Real-World.

In addition to the heat map analysis, we sought to unearth deeper reasons behind the effectiveness of the NC method. Inspired by the t-SNE [38] dimensionality reduction visualization technique, we conducted visualization of the last layer of the fully connected layer feature space on the models trained using both the standard and NC methods, respectively. For illustrative clarity, we employed a soybean dataset containing only three disease categories, enabling direct 3D spatial visualization of feature points. As observed in Figure 10, the model trained using the NC method exhibited greater inter-category distance and closer intra-category proximity in the feature space, indicative of superior classification performance.

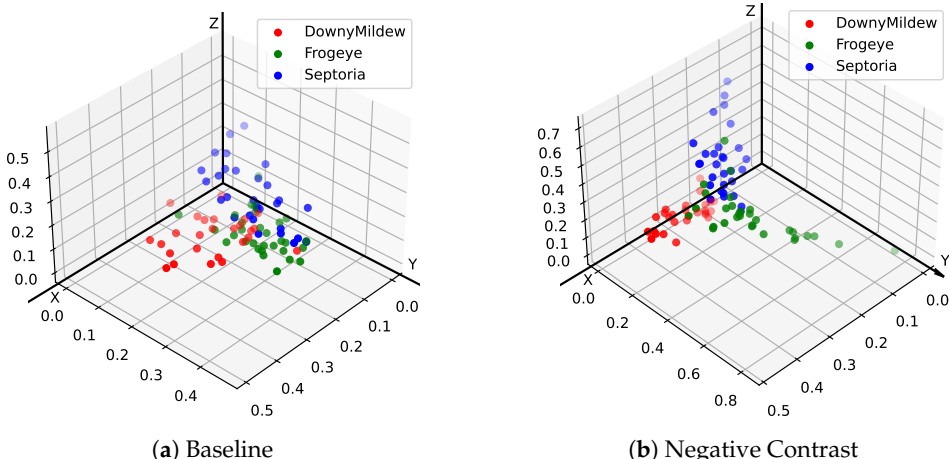

(**a**) Baseline        (**b**) Negative Contrast

**Figure 10.** Visualization of the last layer of features on the soybean test set.

## 4. Discussion

The capability to accurately diagnose crop diseases fundamentally relies on the discernment of features such as the color, shape, and area of the disease spot. This observation serves as the starting point for 'Negative Contrast'. We thus propose an approach akin to fine-grained visual classification to address this problem. Our review of numerous recent results on fine-grained visual classification [39–44] revealed that these models often tend to be overly complex. Older mobile devices, possessing merely one-tenth of the computational power of current state-of-the-art computing devices, struggle to run these models, not to

mention a variety of handheld devices in actual use. The unwieldy size of these models thus poses a significant challenge to their deployment in crop disease classification. This highlights an urgent need for more suitable lightweight models for crop disease classification. Therefore, we did not test large parameter neural network models, such as DenseNet [45], ResNest [46] and Swin Transformer [47], nor did we test transfer models like [48] that have strict requirements on the quantity of the training set. Accordingly, we specifically tested two lightweight models, MobileNetV2 and ShuffleNetV2, in our experiments.

Further experiments suggested that given a training set with highly representative and characteristic disease samples, achieving high model accuracy does not necessarily require a large quantity of samples. For each of the three typical soybean diseases, training with differently proportioned cuts demonstrated that even with the minimal required proportion of images (five percent of the original training set), which equates to only a handful of images for each disease, we could still attain satisfactory classification accuracy on Plant Real-World with negative contrast. This observation underscores the potential of a representative training set to significantly reduce the sample size required for the training set—an invaluable asset in crop disease tasks.

However, negative contrast is not without its limitations. It is somewhat dependent on the dataset for enhancing model performance and tends to rely more on the dataset than the model. As observed in Table 3, while the improvement was smallest for the corn dataset and most significant for the soybean dataset, the enhancement performance of each model under the same dataset was relatively consistent. Moreover, the hand-supervised augmentation method can be resource-intensive with larger data samples compared to [49–53]. Furthermore, when disease areas are irregular and intertwined with the background, the entire disease area, including the background, has to be removed, making the accuracy improvement of this technique less pronounced for certain crop diseases.

Despite these limitations, we envision negative contrast as a potent image augmentation technique. It brings about a significant performance improvement for a variety of models across diverse datasets. The simplicity, effectiveness, and minimal model structure alterations required render it potentially extendable to other domains like medical imaging and anomaly detection. In addition, further performance improvements can be achieved by integrating with the popular object detection model, YoloV4 [54].

Finally, several intriguing questions merit further investigation. For instance, it remains to be explored how drastically the sample size can be reduced using negative contrast without compromising accuracy, or the necessity for a theoretical validation of the effectiveness of such an approach, or even the potential of this technique in enhancing model performance in target detection or segmentation.

## 5. Conclusions

We present "Plant Real-World", a unique real-world dataset encompassing four diverse crops, serving as a valuable resource for further studies in the field of agricultural disease detection. Our proposed "health augmentation" methodology effectively exploits healthy samples, often overlooked in the literature, to suppress the background learning of the network and notably enhances its generalization capacity for crop disease classification tasks. We further introduce a novel image augmentation technique, "negative contrast". In scenarios of sample scarcity, negative contrast can considerably bolster the model's testing accuracy, thereby making a significant contribution to practical real-world applications. Our work lays the groundwork for future explorations into effective utilization of healthy samples and offers promising strategies for tackling the challenge of a limited sample availability.

**Author Contributions:** Conceptualization, J.L. and Z.Y.; method design, J.L.; code, J.L.; validation, J.L., D.L. and Z.Y.; formal analysis, J.L.; investigation, J.L. and Y.Z.; resources, Y.Z.; data collection, J.L.; writing—original draft preparation, J.L.; writing—review and editing, J.L.; visualization, J.L.; supervision, Y.Z. and Z.Y.; project administration, J.L.; funding acquisition, Z.Y. All authors have read and agreed to the published version of the manuscript.

**Funding:** This research received no external funding.

**Institutional Review Board Statement:** Not applicable.

**Data Availability Statement:** Our dataset and codes are accessible at https://www.kaggle.com/datasets/w970704112/corn-wheat-rice-soybean and https://github.com/hiter0/contrastaug, accessed on 24 June 2023.

**Conflicts of Interest:** The authors declare that they have no known competing financial interest or personal relationships that could have appeared to influence the work reported in this paper.

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
