# Peer review of "Negative Contrast: A Simple and Efficient Image Augmentation Method in Crop Disease Classification"

_agriculture, doi:10.3390/agriculture13071461_

Round 1

Reviewer 1 Report

Title: Negative Contrast: A Simple And Efficient Image 4 Augmentation Method In Crop Disease Classification

This work needs a major revision and the following points must be addressed before the acceptance of this paper

1.           This work needs the following improvements:

a.           The introduction needs to be improved considerably. Include the major contributions of this work and why more plants are considered? Please justify.

b.           Move Table 1 to Literature review. Also give the model for deep learning used in Table 1. Ex. ResNet18, ResNet50, … etc.

c.           This section needs to be improved considerably.

d.           There is no dedicated section for Materials and methods. It is very difficult to understand your whole work with out how it is performed.

e.           The characteristics/symptoms for each disease taken for the analysis should be given in detail

2.           The deep learning results present only the GradCam like result. No information regarding the training loss, accuracy and the confusion matrix. Also, the information regarding total images considered, training-testing split, number of classes are missing.

3.           Image augmentation is a common task in deep-learning and hence, Figure 7 can be removed.

4.           The result section and conclusion need to be improved. The present version of the paper is very limited and it needs to be improved.

5. Keep the result table under every figure so that it will be helpful for the easier understanding of the performance of the algorithms used in this study

Minor Comments

1. Rewrite this statement – “plant disease 83 leaves for image classification, and PlantDoc[24], which covers 13 plant species and 17”

2. There is no space after a full stop. “Crop Disease Classification.The rap” correct it

3. The author should be mentioned before the reference.  Correct this statement

may also be a result of the sample distribution itself. [44] showed that the distribution of     210

4. The following data are incomplete

Author Contributions: 322 Funding: 323 Data Availability Statement: 324 Conflicts of Interest:

Language must be checked thoroughly 

Author Response

Thank you for your feedback. In response to your comments, I have made the following revisions to the manuscript:

1.a. I have rewritten the Introduction section of the paper. 1.b. I removed Table 1 and included its content within the relevant work section of the Introduction, where I provide a brief overview of its original content. 1.c, d. I have restructured the manuscript, separating the Materials and Methods from the Results to make it easier for readers to understand the implementation of my novel contributions. 1.e. I have reorganized the statistical data from the original dataset, and in this version, details are provided for each disease category.

2.In the Results section, I have reorganized the comparison results of different image enhancement methods, improvement comparison results of three models, GradCAM visualization result diagrams, and added comparison curves of test accuracy changes during the training process of the three models. In the Materials and Methods section, I have described more information about all images. In this study, due to the multitude of experimental results produced by different models and crops, even if I were to list a certain model or crop, I could not fit the confusion matrix and training loss graphs into the main text. I sincerely hope for your understanding in this matter.

3.The suggested changes have been incorporated into the new version.

4.The new version has undergone comprehensive modifications.

5.After removing Figure 7 from the previous version, I have tried my best to satisfy this suggestion.

Regarding the minor issues in the manuscript: I have conducted a thorough and detailed review of all the English writing issues present and made the necessary corrections. Additionally, missing information has been supplemented where necessary.

Your comments are very much appreciated and I believe they have helped improve the overall quality of the manuscript.

Reviewer 2 Report

The manuscript, “Negative Contrast: A Simple And Efficient Image Augmentation Method In Crop Disease Classification” is a generally interesting paper highlighting an uncommon photo image enhancement method applied to an agricultural context. I accepted the review based on the abstract, and while it is not at all misleading, it does cover an aspect of the field in which I have no experience. However, the methods, if not the models, do translate and thus I can review this paper. 

1: the manuscript needs editing for clarity and precision. Too often the authors rely on casual language use leaving the reader unsure what is being said. Examples, “Unlike distinguishing images such as birds[7], dogs[8], aircraft[9], crop diseases have much smaller inter-class differences, and the number of individual disease samples is sparse, and in many cases it is even necessary to distinguish disease classes by some details.” And “However, datasets from the same distribution tend to overfit our models, and the models may learn features that are not related to the disease, such as the land or the healthy leaves around the disease, etc., thus achieving a fairly high validation accuracy.”  At this point in the paper, the reader has no idea what models or datasets are being discussed. This pattern continues throughout. 

2: Negative contrast is underexplained. The section and figure devoted to this obscures both the replication model and the utility of the tool. Language issue again, “However, the imbalance of the data and the scarcity of some samples may also be a result of the sample distribution itself.“ While it is clear that photoshop is used to create these images, I could not replicate this given the descriptions provided. This is critical for revision. 

3: The models described are new to me: MoblieNetV2, ResNet50 and ShuffleNetV2. The paper should include more description and references to these models. It is possible that the model design influenced the success of the outcome but I cannot be sure given what is presented. 

4: I appreciated the figures and recommend that Figure 6 be redesigned to be larger and more informative. It is the most important figure overall.  It is very small and could use another column for explanation. 

Minor points:

1: I found the name health augmentation confusing in the context of disease identification. It is a confusing phrase focusing on “not-disease” and not augmentation of health or healthy zones per se. I wouldn’t refer to image augmentation, it really is a segmentation model. 

2: The paper has many spacing errors.

The English is acceptable, but it is a bit too casual and lacks precision. 

Author Response

Thank you for your advice. I have made the following modifications in response to the issues raised in your comments:

1.I have completely rewritten the relevant parts of the text. All statements that could potentially confuse readers have been corrected.

2.I have restructured the article, separating the methods from the results, and added some experimental results and necessary explanations. Regarding the name "Negative Contrast", I have provided a new explanation in the introduction and main text, in hopes that readers can understand the core idea of this algorithm.

3.The missing references and meanings of the terms related to these deep learning network models have been supplemented on lines 125 to 141 of the paper.

4.Thank you for your appreciation of this figure. According to your suggestion, I have divided it into two more understandable figures and provided separate explanations for the two parts.

For the minor issues:

1.The term "Health Augmentation" is a summarizing explanation of the method used in the paper. I have slightly modified its description on lines 150 to 157 in the main text, hoping that readers can understand it.

2.Regarding potential English writing issues in the original text, I have conducted a thorough check and comprehensive corrections.

In conclusion, I would like to express my deep gratitude for the time and effort you have invested in reviewing my paper. Your valuable comments and suggestions have significantly contributed to improving the quality of this work. I am looking forward to any further feedback you might have.

Reviewer 3 Report

The work was done on an urgent topic devoted to the development of a system for the classification of plant diseases. The topic is certainly relevant. The authors present an original approach that improves the efficiency of classifying diseased plants. In itself, the approach of controlled image magnification is not new, however, for this task it represents some originality. However, there are a few remarks worth noting.

1. Link to dataset and program code is definitely an advantage. However, it is not clear why the authors provide these links in the annotations to the work. It would be useful to insert these links in a section that describes materials, methods, and data. Also, the link https://www.kaggle.com/datasets/w970704112/corn-wheat-rice-soybean is not working.

2. In the work, 3 algorithms are given as a comparison: ResNet50, MobileNetV2, ShuffleNetV2. Undoubtedly, these algorithms are popular and are actively used in various computer vision tasks. Nevertheless, the quality of the algorithms, despite the improvements made, leaves much to be desired. Perhaps you should be more careful with setting the parameters. More modern algorithms, such as YOLOv5, might improve the performance of the model.

Despite this, the work as a whole is of research interest and can be published after minor modifications.

Author Response

Thank you for your insightful suggestions. I have made the following amendments based on your comments:

1.I have adjusted the permissions on the dataset link and it is now accessible. Regarding the placement of these links, I've reviewed many previously published papers and found that including the link in the abstract is a common practice. I haven't been able to find a specific reference on embedding these links within the main body of the paper. If possible, could you please provide further guidance on where these links should be placed or what different effect they might have when placed elsewhere?

2.I have included a comparative experimental result graph, Figure 8, which provides a detailed analysis showing that the performance changes are brought about by Negative Contrast, not the model itself. As for the YoloV5 algorithm, it is a method applied in the field of object detection. However, our focus is on image classification tasks, therefore, this algorithm is not within our scope of comparison.

I hope these amendments address your concerns and look forward to your further feedback.

Round 2

Reviewer 1 Report

Accept

Accept